# Chronic central retinal artery occlusion: Clinical manifestations, ocular neovascular complications, and risk of stroke

**Jae Ryong Song[1,2], Se Joon Woo [2]***

1 Department of Ophthalmology, Seoul National University College of Medicine, Seoul National University Hospital, Seoul, Republic of Korea, 2 Department of Ophthalmology, Seoul National University College of Medicine, Seoul National University Bundang Hospital, Seongnam, Republic of Korea

* sejoon1@snu.ac.kr

## Abstract

Central retinal artery occlusion (CRAO) is a vision-threatening emergency that can lead to chronic ischemic complications. This retrospective cohort study investigated clinical characteristics, neovascular complications, and stroke risk in patients with chronic CRAO defined as the post-acute phase state characterized by FA-confirmed persistent retinal ischemia ≥1 month after acute onset. We reviewed medical records of 570 eyes initially diagnosed with acute non-arteritic CRAO between September 2003 and December 2024, with 289 eyes meeting inclusion criteria. Chronic CRAO was defined as persistent prolonged arm-to-retina or arteriovenous transit times on fluorescein angiography performed >1 month after symptom onset, and further classified as delayed perfusion or nonperfusion based on venous filling patterns. Among 289 patients, 215 eyes (74.4%) had acute CRAO with spontaneous reperfusion, while 74 eyes (25.6%) had chronic CRAO, including 55 eyes (74.3%) with delayed perfusion and 19 eyes (25.7%) with nonperfusion. Chronic CRAO patients were older with higher prevalence of systemic comorbidities. Previous stroke (43.2% vs. 26.0%, p = 0.009) and concurrent stroke (20.3% vs. 12.1%, p = 0.122) were more common in chronic CRAO patients compared to acute CRAO. Neovascularization of the iris occurred exclusively in chronic CRAO (32.4% vs. 0.0%, p < 0.001), with the nonperfusion group showing significantly higher rates of neovascularization of the iris (90% vs. 13%, p < 0.001) and neovascular glaucoma (68% vs. 9%, p < 0.001) compared to delayed perfusion group. Kaplan-Meier analysis revealed earlier onset of neovascularization in nonperfusion cases (median 1.6 months) with a 20-fold increased hazard ratio. Approximately one quarter of acute non-arteritic CRAO patients progress to chronic CRAO, which is associated with increased stroke risk and ocular neovascularization, particularly when retinal arterial nonperfusion persists. These findings emphasize the importance of fluorescein angiography monitoring and systemic evaluation in CRAO management.

**Data availability statement:** The datasets generated and/or analyzed during the current study contain sensitive patient information and cannot be shared publicly due to patient privacy and confidentiality restrictions under Korean personal information protection laws. Data are available from the corresponding author (contact: sejoon1@snu.ac.kr) and the Institutional Review Board of Seoul National University Bundang Hospital (email: snubhirb@gmail.com; Tel: +82-31-787-8801) for researchers who meet the criteria for access to confidential data, upon reasonable request and following approval of a data sharing agreement. Data will be maintained securely at Seoul National University Bundang Hospital for the duration required by institutional and regulatory policies.

**Funding:** This work was supported by the National Research Foundation of Korea (NRF) grant funded by the Korean government (MSICT) (RS-2023-00248480) and by a grant of Korean ARPA-H Project through the Korean Health Industry Development Institute (KHIDI), funded by the Ministry of Health and Welfare, Republic of Korea (grant number: RS-2024-00512384). The funding organization had no role in the design or conduct of this study.

**Competing interests:** The authors have declared that no competing interests exist.

## Introduction

Central Retinal Artery Occlusion (CRAO) is a vision-threatening ophthalmologic emergency characterized by sudden interruption of blood flow to the retina, resulting in acute ischemia and significant vision impairment [1]. Although CRAO has a relatively low overall incidence, its clinical significance lies in the high risk of irreversible vision loss and subsequent ocular complications [2]. While the acute emergency phase of CRAO is the focus of most clinical attention, the post-acute phase presents distinct clinical challenges. Although most CRAO cases result in permanent visual loss due to acute ischemia, some patients achieve reperfusion on follow-up fluorescein angiography(FA) while others exhibit persistent perfusion impairment, with prior studies demonstrating that persistent nonperfusion is strongly associated with neovascular complications [3].

Neovascularization of the iris (NVI) represents a severe complication of chronic ischemic ocular diseases, including proliferative diabetic retinopathy, ocular ischemic syndrome, and central retinal vein occlusion [4]. Driven by increased vascular endothelial growth factor (VEGF) expression in hypoxic retinal tissue, NVI can extend into the trabecular meshwork, leading to neovascular glaucoma (NVG). This complication frequently manifests as dramatically elevated intraocular pressure and may result in intractable pain or permanent vision loss [5]. Previous studies have reported widely varying incidences of NVI in acute CRAO, with rates ranging from 2.5% to 31.6% over subsequent weeks to months [2,3,6–8]. Despite documentation of NVI and NVG development in CRAO patients, the precise incidence of these complications remains debated [3,6,8,9].

The limited consensus regarding the occurrence of neovascular complications in CRAO raises concerns about optimal patient management and prognosis. Specifically, uncertainty persists regarding appropriate monitoring protocols, the role of prophylactic interventions such as pan-retinal photocoagulation in preventing NVI/NVG, and clinical features that might identify high-risk individuals [10]. Given that chronic CRAO can significantly diminish vision-related quality of life, addressing these knowledge gaps is paramount.

Patients with CRAO have a higher chance of ischemic stroke, with a self-controlled case series in the Korean population demonstrating a 14-fold increase in stroke incidence within the first 30 days following CRAO onset [11]. Moreover a single-center retrospective cohort reported an 8.6% one-year stroke event rate, particularly ipsilateral stroke associated with large artery atherosclerosis [12]. However, to date, no specific CRAO subtype has been identified as being associated with differential stroke risk.

In this study, we define "Chronic CRAO" as the post-acute phase (≥1 month after onset) characterized by FA-confirmed persistent retinal ischemia, distinct from the acute emergency presentation. We aimed to clarify the clinical features, risk factors, and ocular outcomes of chronic CRAO, with a particular focus on NVI and NVG. We also assessed the prevalence of subsequent stroke after CRAO onset according to CRAO types. We classified chronic CRAO cases according to their fluorescein angiography (FA) perfusion status, hypothesizing that those with persistent nonperfusion

would exhibit more frequent neovascular complications and worse visual prognosis. Through this approach, we sought to enhance risk stratification for patients with chronic CRAO and offer evidence-based guidance for prophylactic and therapeutic interventions in high-risk subgroups.

## Method

This study was approved by the Institutional Review Board of Seoul National University Bundang Hospital (IRB No. B-2410-931-104) and adhered to the tenets of the Declaration of Helsinki. The requirement for written informed consent was waived due to the retrospective nature of the study. We conducted a retrospective review of medical records for all patients diagnosed with chronic central retinal artery occlusion (CRAO) at Seoul National University Bundang Hospital between September 2003 and December 2024.

### Study population

We identified acute CRAO patients through a comprehensive review of electronic medical records. Initial screening involved electronic medical record review using ICD-10 codes (H34.10-H34.13), followed by clinical validation where all cases were reviewed by board-certified retinal specialists (S.J.W. and J.R.S.). Imaging confirmation required systematic review of fundus photography confirming cherry-red spot and retinal whitening, fluorescein angiography demonstrating retinal arterial occlusion, and OCT confirming retinal structural changes consistent with CRAO. Exclusion criteria were: (1) follow-up periods of less than one month, (2) concurrent central retinal vein occlusion (CRVO) or proliferative diabetic retinopathy (PDR), (3) iatrogenic or arteritic CRAO, and (4) absence of follow-up fluorescein angiography (FA) (5) unclear CRAO diagnosis based on clinical history or fundus findings. Patients with CRVO or PDR were excluded to avoid confounding factors, as these conditions independently increase the risk of neovascularization of the iris (NVI) and neovascular glaucoma (NVG) [4]. A total of 570 patients diagnosed with central retinal artery occlusion (CRAO) were initially considered, and 289 patients met the eligibility criteria for analysis (Fig 1).

### Clinical evaluation and treatment

At presentation, all patients underwent a comprehensive ophthalmologic examination, including visual acuity testing, slit-lamp microscopy, intraocular pressure (IOP) measurement using Goldmann applanation tonometry, and fundus examination. Gonioscopy was not routinely performed in all patients at baseline but was selectively conducted during follow-up in cases with suspected neovascularization or elevated IOP. CRAO diagnosis was confirmed based on characteristic clinical findings, such as acute severe vision loss, retinal whitening, and a cherry-red spot at the posterior pole. Treatment options included intra-arterial thrombolysis for eligible patients who met the treatment criteria [13] and agreed to the treatment. Other patients received conservative management, which involved ocular massage, anterior chamber paracentesis, hyperbaric oxygen therapy, and anticoagulation.

### Follow-up and complications assessment

Patients were followed at 1- to 3-month intervals and monitored for NVI development (abnormal iris neovascularization on slit-lamp examination) and NVG (IOP greater than 21 mmHg with concurrent NVI). Retinal circulation was evaluated using fluorescein angiography, with reperfusion defined as an improvement in arm-to-retina time and arteriovenous passage time. If FA demonstrated persistent perfusion impairment, serial FA examinations were conducted thereafter to confirm sustained impairment and to monitor disease courses. "Chronic CRAO"(CRAO with persistent retinal ischemia) was defined as cases where fluorescein angiography performed more than one month after diagnosis showed persistent delay in arm-to-retina time (>20 seconds) and arteriovenous transit time (>11 seconds). The 1-month cutoff was chosen based on established literature showing that most acute retinal changes in CRAO resolve within 2–4 weeks, with persistent findings beyond this time point representing chronic ischemic sequelae rather than ongoing acute changes [14].

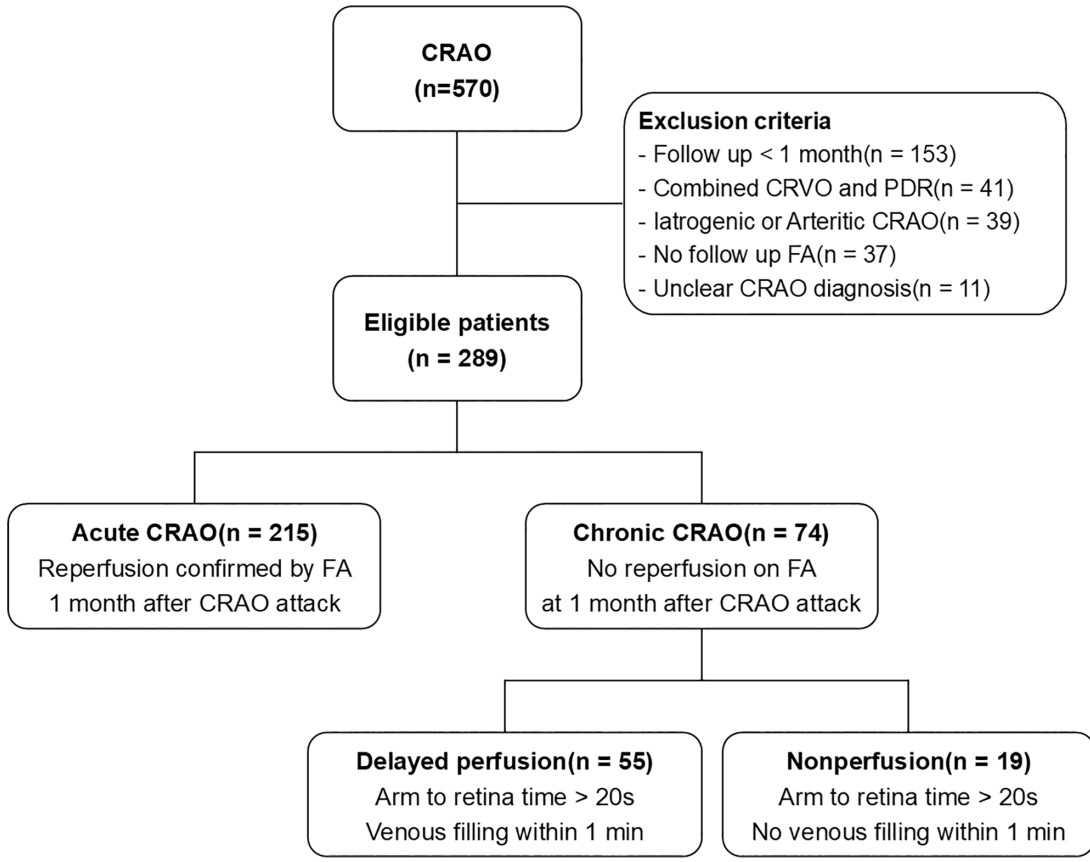

**Fig 1. Flow chart of study design and exclusion criteria.** A total of 570 CRAO cases were reviewed and 289 eyes met the inclusion criteria after excluding cases with short follow-up, confounding retinal diseases, or missing FA data. (CRAO = central retinal artery occlusion; CRVO = central retinal vein occlusion; PDR = proliferative diabetic retinopathy; FA = fluorescein angiography).

This timeframe also aligns with clinical follow-up protocols. Given that all patients who developed NVI in previous studies exhibited nonperfusion on fluorescein angiography [3], chronic CRAO cases were further categorized into two groups: the Delayed Perfusion group, where arm-to-retina time was prolonged (>20s) but venous filling was observed within one minute, and the Nonperfusion group, where perfusion did not occur even after one minute (Fig 2).

## Systemic evaluation and strokes

All patients underwent a comprehensive systemic evaluation by neurologists or cardiologists to assess for atherosclerotic diseases, including cerebrovascular and cardiovascular conditions, diabetes mellitus, hyperlipidemia, hypertension, and carotid artery stenosis. Stroke events were assessed using brain imaging (MRI and/or CT) performed at the time of CRAO diagnosis or during follow-up. We classified stroke events into three distinct categories based on temporal relationships to CRAO onset. Previous stroke was defined by patient history or old infarction on brain MRI. Concurrent stroke represented acute cerebral infarctions detected on MRI performed at the time of CRAO diagnosis, indicating simultaneous vascular events within the same or different vascular territories. Incident stroke included new cerebral infarctions detected on follow-up imaging performed after CRAO diagnosis. Carotid patency was evaluated using Doppler ultrasonography, brain computed tomography angiography (CTA), or brain magnetic resonance angiography (MRA)

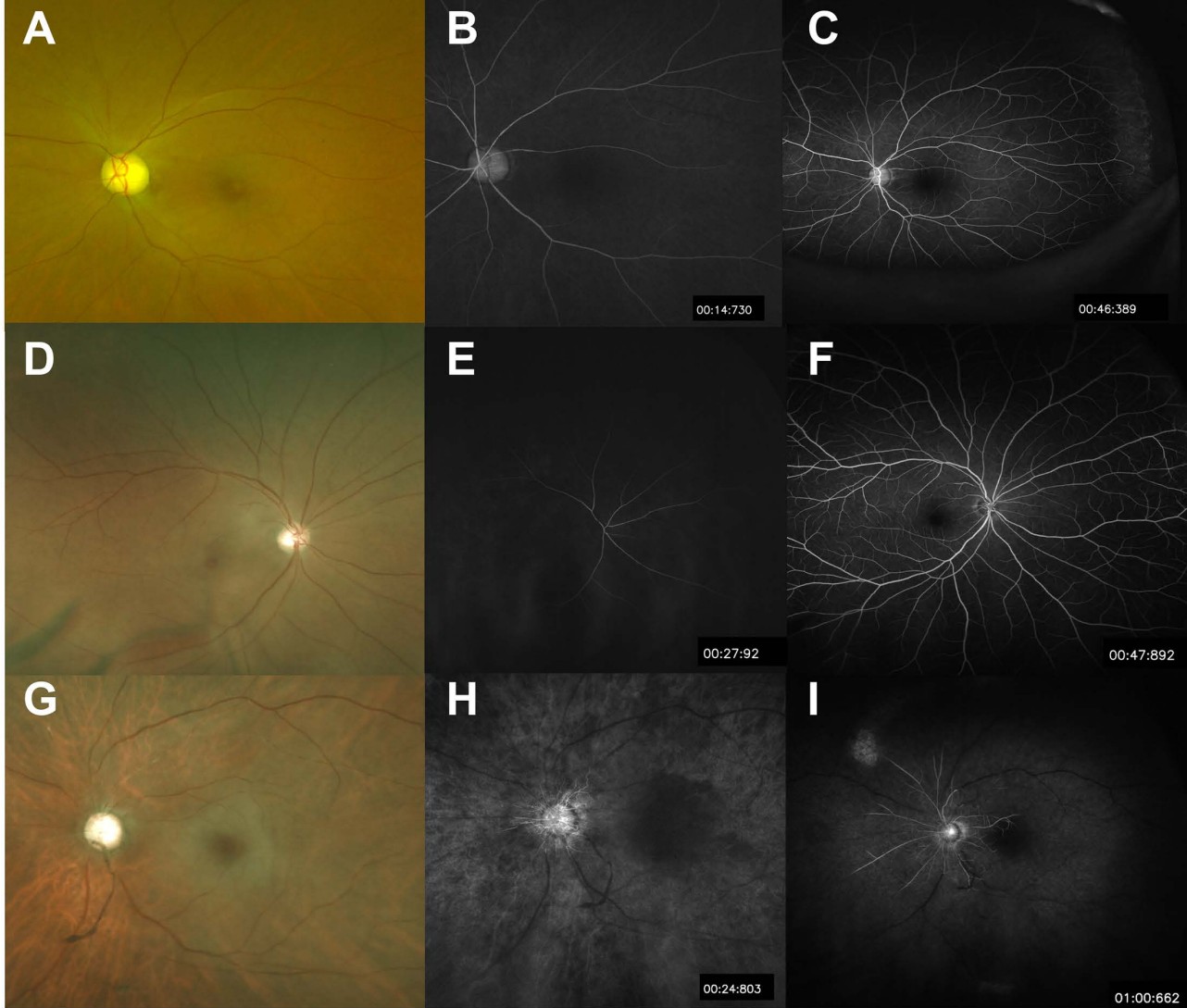

**Fig 2. Classification of acute and chronic CRAO based on fluorescein angiography (FA) findings at 1 month after CRAO attack with representative images.** This figure illustrates the classification of CRAO based on FA findings at 1 month after CRAO attack. Acute CRAO demonstrates normal arm-to-retina circulation time and arteriovenous transit time **(A–C)**. Chronic CRAO is further divided into two subtypes: chronic CRAO with delayed perfusion and chronic CRAO with nonperfusion. In chronic CRAO with delayed perfusion, the arm-to-retina circulation time is prolonged to more than 20 seconds, but venous filling is observed within 1 minute **(D–F)**. In contrast, chronic CRAO with nonperfusion shows no venous filling even after 1 minute (G–I).

and classified according to the NASCET criteria, which defines stenosis based on the ratio of the residual lumen to the normal distal lumen diameter [15].

## Statistical analysis

Statistical analyses were performed using SPSS software (version 25.0, IBM Corp., Chicago, IL, USA). We performed comparisons between acute and chronic CRAO, as well as between the Delayed Perfusion and Nonperfusion groups, using independent t-tests or Mann-Whitney U tests for continuous variables and chi-square tests or Fisher's exact tests

for categorical variables. Kaplan-Meier survival curves were generated to assess the time to NVI occurrence in the Delayed Perfusion and Nonperfusion groups, with differences evaluated using the log-rank test. A p-value of less than 0.05 was considered statistically significant.

## Results

Among the 289 eligible patients, 215 (74.4%) were classified as acute CRAO, showing partial or complete spontaneous reperfusion on follow-up FA, while 74 (25.6%) were classified as chronic CRAO, defined by persistent perfusion impairment beyond one month based on serial FA evaluations. Within the chronic CRAO group, 55 patients (74.3%) exhibited delayed perfusion, whereas 19 patients (25.7%) demonstrated nonperfusion (Fig 1).

### Demographic and clinical characteristics in CRAO

A comparative analysis of clinical and retinal structural characteristics among CRAO subtypes is summarized in Table 1. Patients with chronic CRAO were significantly older and exhibited higher prevalence of systemic comorbidities such as diabetes mellitus, hypertension and previous stroke compared to those with acute CRAO. In terms of retinal perfusion parameters, patients with chronic CRAO had longer initial arm-to-retina times (p = 0.003) and follow-up arm-to-retina times (p < 0.001) compared to acute CRAO, indicating persistent impaired retinal circulation. Visual outcomes were significantly worse in chronic CRAO, both in baseline and final BCVA LogMAR values.

Moreover, patients in the nonperfusion group were significantly older than those in the delayed perfusion group. However, other systemic comorbidities, including diabetes, hypertension, dyslipidemia, and cardiovascular disease, were not significantly different between the two groups. Moreover, Patients in the nonperfusion group had significantly worse final BCVA compared to the delayed perfusion group, indicating more severe visual impairment.

### Ocular neovascular complications in CRAO

Neovascularization of the iris (NVI) was observed exclusively in the chronic group, suggesting a higher risk of ischemic complications. Within the chronic CRAO group, neovascularization of the iris (NVI, 89.5% vs. 12.7%, p < 0.001) and neovascular glaucoma (NVG, 68.4% vs. 9.1%, p < 0.001) were significantly more prevalent in the nonperfusion group, highlighting a greater risk of ischemic complications in these patients (Table 1).

A comparison of NVI onset timing and treatment patterns between delayed perfusion and nonperfusion CRAO patients is shown in Table 2. The mean time from symptom onset to NVI development was significantly shorter in the nonperfusion group compared to the delayed perfusion group(p = 0.042). Regarding treatment interventions, pan-retinal photocoagulation (PRP) and anti-VEGF therapy was performed more frequently in the nonperfusion group, though the difference was not statistically significant.

Fig 3 illustrates the Kaplan-Meier survival curves comparing the cumulative incidence of NVI between patients with delayed perfusion and nonperfusion CRAO. The log-rank test demonstrated a statistically significant difference between the two groups (p < 0.001), indicating that patients with nonperfusion CRAO had a significantly higher and more rapid risk of developing NVI compared to those with delayed perfusion CRAO. In addition, log-rank hazard ratio analysis revealed that patients with nonperfusion CRAO had a 20.17-fold higher risk (95% CI: 6.14–66.24) of developing NVI compared to patients with delayed perfusion CRAO, confirming the significantly increased risk in this patient subgroup.

### Systemic complications in CRAO – stroke risk and carotid stenosis

The proportion of patients with previous stroke was significantly higher in the chronic CRAO group compared to the acute CRAO group. Specifically, previous stroke increased from 26.5% in acute CRAO to 40.0% in delayed perfusion (p = 0.0498) and 52.6% in nonperfusion cases (p = 0.0158 vs. acute CRAO). When considering overall stroke burden

**Table 1. Comparison of clinical characteristics and retinal structural findings among CRAO subtypes.**

| Characteristics | Total (n = 289) | Acute CRAO (n = 215, 74.4%) | Chronic CRAO (n = 74, 25.6%) | P value | Chronic CRAO with delayed perfusion (n = 55, 74.3%) | Chronic CRAO with nonperfusion (n = 19, 25.7%) | P value |
|---|---|---|---|---|---|---|---|
| Age | 63.7 ± 15.2 | 62.0 ± 16.2 | 68.9 ± 10.3 | <0.001* | 67.2 ± 9.5 | 73.6 ± 11.1 | 0.013* |
| Male, n(%) | 177(61.2) | 122(56.7) | 55(74.3) | 0.008† | 41(74.5) | 14(73.7) | 1.000† |
| DM, n(%) | 57(19.7) | 33(15.3) | 24(32.4) | 0.002† | 15(27.3) | 9(47.4) | 0.155† |
| HTN, n(%) | 156(54.0) | 105(48.8) | 51(68.9) | 0.003† | 37(67.3) | 14(73.7) | 0.776† |
| DL, n(%) | 69(23.9) | 44(20.5) | 25(33.8) | 0.027† | 21(38.2) | 4(21.1) | 0.261† |
| Previous stroke, n(%) | 88 (30.4) | 56 (26.0) | 32 (43.2) | 0.009† | 22 (40.0) | 10 (52.6) | 0.345† |
| Concurrent stroke, n(%) | 41(14.2) | 26(12.1) | 15(20.3) | 0.122† | 11(20.0) | 4(21.1) | 1.000† |
| Incident stroke, n(%) | 9(3.1) | 6(2.8) | 3(4.1) | 0.879† | 2(3.6) | 1(5.3) | 1.000† |
| Time from onset to stroke(months) | 8.8 ± 26.5 (range 0.0–125.0) | 11.1 ± 31.1 (range 0.0–125.0) | 5.1 ± 16.9 (range 0.0–70.0) | 0.944* | 5.5 ± 19.4 (range 0.0–70.0) | 4.2 ± 9.4 (range 0.0–21.0) | 0.879* |
| Cardiovascular disease, n(%) | 40(13.8) | 27(12.6) | 13(17.6) | 0.329† | 10(18.2) | 3(15.8) | 1.000† |
| Time from symptom onset to visit(hours) | 167.0 ± 524.9 | 181.3 ± 583.4 | 126.3 ± 300.9 | 0.439* | 100.4 ± 279.4 | 201.4 ± 353.4 | 0.158* |
| IAT, n(%) | 107(37.0) | 80(37.2) | 27(36.5) | 1.000† | 23(41.8) | 4(21.2) | 0.166† |
| Baseline BCVA(LogMAR) | 2.7 ± 1.0 | 2.6 ± 1.0 | 2.9 ± 1.0 | 0.048* | 2.9 ± 1.0 | 2.8 ± 1.0 | 0.773* |
| Baseline CMT(μm) | 331.1 ± 121.0 | 330.7 ± 123.9 | 332.4 ± 114.0 | 0.945* | 332.7 ± 125.2 | 331.5 ± 79.9 | 0.752* |
| Initial Arm to Retina Time (ART) | 21.5 ± 10.0 | 20.3 ± 10.0 | 25.0 ± 9.2 | 0.003* | 24.6 ± 7.8 | 26.2 ± 12.7 | 0.924* |
| Follow up ART | 18.0 ± 6.4 | 15.7 ± 4.8 | 23.3 ± 6.5 | <0.001* | 22.7 ± 6.2 | 25.6 ± 7.1 | 0.124* |
| Final BCVA(LogMAR) | 2.3 ± 1.3 | 2.2 ± 1.2 | 2.8 ± 1.3 | <0.001* | 2.7 ± 1.3 | 3.3 ± 1.3 | 0.017* |
| Final CMT(μm) | 230.3 ± 51.7 | 229.8 ± 41.9 | 231.8 ± 76.9 | 0.813* | 232.8 ± 83.0 | 226.9 ± 40.4 | 0.757* |
| NVI, n(%) | 24(8.3) | 0(0.0) | 24(32.4) | <0.001† | 7(12.7) | 17(89.5) | <0.001† |
| NVG, n(%) | 18(6.2) | 0(0.0) | 18(24.3) | <0.001† | 5(9.1) | 13(68.4) | <0.001† |

Values are presented as mean ± standard deviation or number (%).

* p-value was calculated by independent T-test or Mann-Whitney U test.

† p-value was calculated by Chi-Square test or Fisher exact test.

CRAO = central retinal artery occlusion; DM = diabetes mellitus; HTN = hypertension; DL = dyslipidemia; IAT = intra-arterial thrombolysis; CMT = central macular thickness; BCVA = best-corrected visual acuity; LogMAR = logarithmic minimum angle of resolution; NVI = neovascularization of iris, NVG = neovascular glaucoma.

(including previous, concurrent, and incident stroke), there was a significant stepwise increase across CRAO types: 34.0% in acute CRAO, 54.5% in delayed perfusion (p = 0.005 vs. acute), and 63.2% in nonperfusion cases (p = 0.011 vs. acute). Concurrent and incident stroke events also showed a stepwise increase across groups, though these differences were not statistically significant (Table 1, Fig 4).

Fig 5 presents the distribution of carotid artery stenosis severity across different CRAO subgroups. Carotid stenosis patterns differed significantly between acute and chronic CRAO groups (p < 0.001), with both chronic CRAO subtypes demonstrating markedly worse carotid stenosis distributions compared to acute CRAO. The acute CRAO group showed

**Table 2. Comparison of NVI Onset and treatment patterns between chronic CRAO patients with delayed perfusion and those with nonperfusion.**

| Characteristics | Chronic CRAO patients with NVI(n = 24) | Delayed perfusion with NVI(n = 7) | Nonperfusion with NVI(n = 17) | P value |
|---|---|---|---|---|
| Mean time from onset to NVI(weeks) | 16.7 ± 20.5 | 34.3 ± 31.4 | 9.5 ± 6.5 | 0.042* |
| Median time from onset to NVI(weeks) | 8.8 (3.0 to 82.4) | 17.7 (4.4 to 82.4) | 5.9 (3.0 to 22.0) | – |
| NVG, n(%) | 18(75.0) | 5(71.4) | 13(76.5) | 1.000† |
| PRP, n(%) | 14(58.3) | 2(28.6) | 12(70.6) | 0.085† |
| Anti-VEGF, n(%) | 21(87.5) | 5(71.4) | 16(94.1) | 0.194† |
| Ahmed implant operation, n(%) | 3(12.5) | 1(14.3) | 2(11.8) | 1.000† |

Values are presented as mean ± standard deviation or number (%).

* p-value was calculated by the Mann-Whitney test.

† p-value was calculated by the Fisher Exact test.

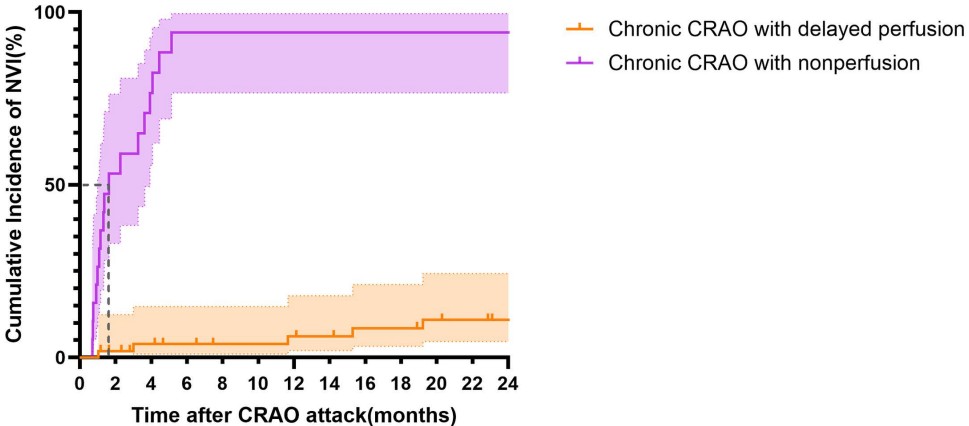

**Fig 3. Cumulative Incidence of Neovascularization of the Iris (NVI) in Chronic CRAO Patients Based on Fluorescein Angiography (FA) Classification.** Kaplan-Meier curves showing the cumulative incidence of NVI in patients with chronic CRAO, categorized by FA findings at 1-month after CRAO attack. Chronic CRAO with nonperfusion(purple) had a significantly higher incidence of NVI compared to the chronic CRAO with delayed perfusion (orange) (log-rank test, P < 0.001). Median time to NVI onset in the nonperfusion group was 1.6 months. Shaded areas represent the 95% confidence intervals (CI) for each group.

predominantly mild or no carotid disease, while both chronic CRAO subtypes exhibited substantial shifts toward more advanced carotid stenosis and complete occlusion. Moreover, while the nonperfusion group showed a trend toward higher proportions of severe stenosis and occlusion, the overall carotid stenosis distributions between the two chronic CRAO subtypes were not statistically different (delayed perfusion vs. nonperfusion, p = 0.392), suggesting that both persistent ischemia groups share comparable underlying advanced vascular disease burden.

When examining patients who developed neovascular complications, carotid stenosis patterns showed no significant differences between chronic CRAO subtypes (p = 0.181), though both groups demonstrated high rates of severe carotid disease. This observation suggests that neovascular complications can develop even in the presence of varying degrees of carotid pathology, indicating that factors beyond large vessel stenosis may contribute to persistent retinal ischemia.

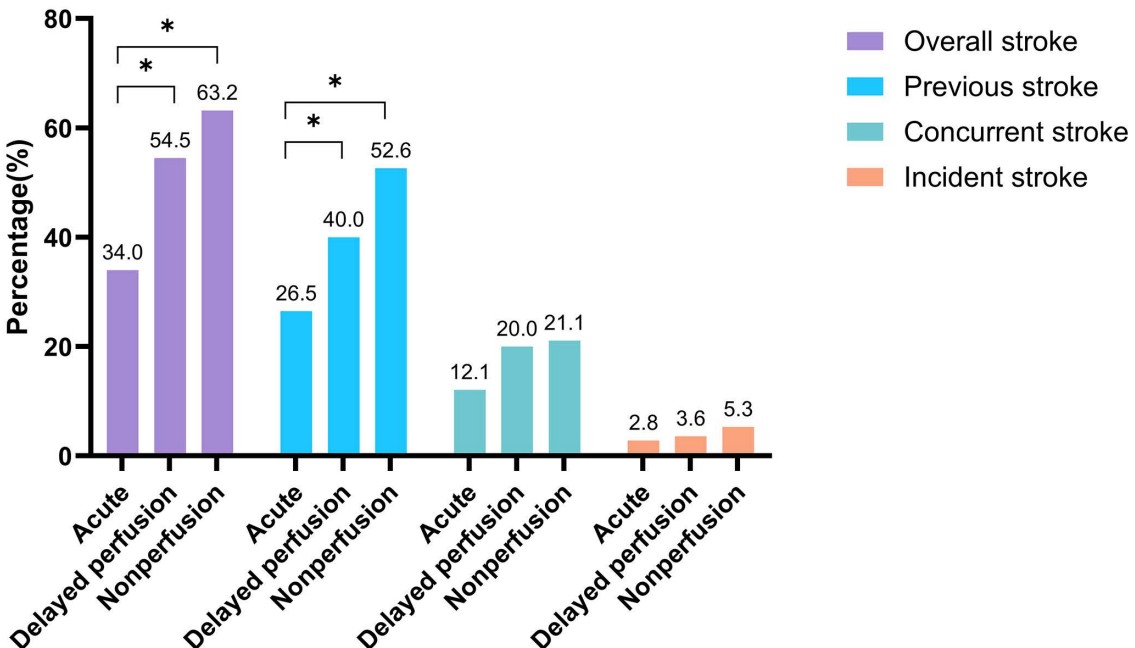

**Fig 4. Incidence of stroke according to the chronicity and perfusion status.** Incidence of overall stroke incidence, previous stroke, concurrent stroke, and incident stroke according to the chronicity (acute vs chronic) and perfusion status (delayed perfusion vs nonperfusion). Among the subgroups, the chronic CRAO with nonperfusion group showed the highest percentage of overall stroke incidence and previous stroke, while concurrent and incident stroke rates did not differ significantly across groups.

## Discussion

This study introduces the novel concept of chronic CRAO and demonstrates its clinical significance with high stroke risk, ocular neovascular complications and worse visual prognosis, suggesting the importance of classifying patients using FA-based retinal perfusion status. Moreover, among chronic CRAO patients, the nonperfusion group exhibited significantly worse visual outcomes and a markedly higher risk of developing NVI and NVG compared to the delayed perfusion group. Our results indicate that chronic CRAO patients, especially in the nonperfusion group, require close monitoring including FA.

The pathophysiological mechanism underlying neovascularization focuses on retinal ischemia-driven angiogenic factor release. Prolonged hypoxia in the retina upregulates vascular endothelial growth factor (VEGF) and other mediators, which diffuse into the anterior segment and promote iris neovascularization. Studies have consistently shown elevated VEGF levels in the ocular fluids of patients with ischemic retinal disorders, supporting its role in ocular neovascularization [16]. The pronounced differences in clinical outcomes between acute and chronic CRAO, as well as between the delayed perfusion and nonperfusion groups, would be largely attributable to this process. Chronic retinal ischemia, particularly in cases of nonperfusion, exacerbates hypoxia and further promotes VEGF upregulation. Moreover, delayed perfusion group maintains venous filling despite prolonged arm to retina time, indicating less severe ischemia and reduced VEGF elevation compared to nonperfusion group. This difference in perfusion status would likely explain the significantly higher incidence of NVI and NVG in the nonperfusion group compared to the delayed perfusion group. Additionally, NVI in the delayed perfusion group tended to develop later than in the nonperfusion group, suggesting that residual retinal circulation may slow the progression of neovascular complications.

Our findings align with several previous studies documenting the relationship between CRAO and subsequent neovascular complications. Our previous study reported a 10.9% incidence of NVI following acute CRAO, with neovascularization

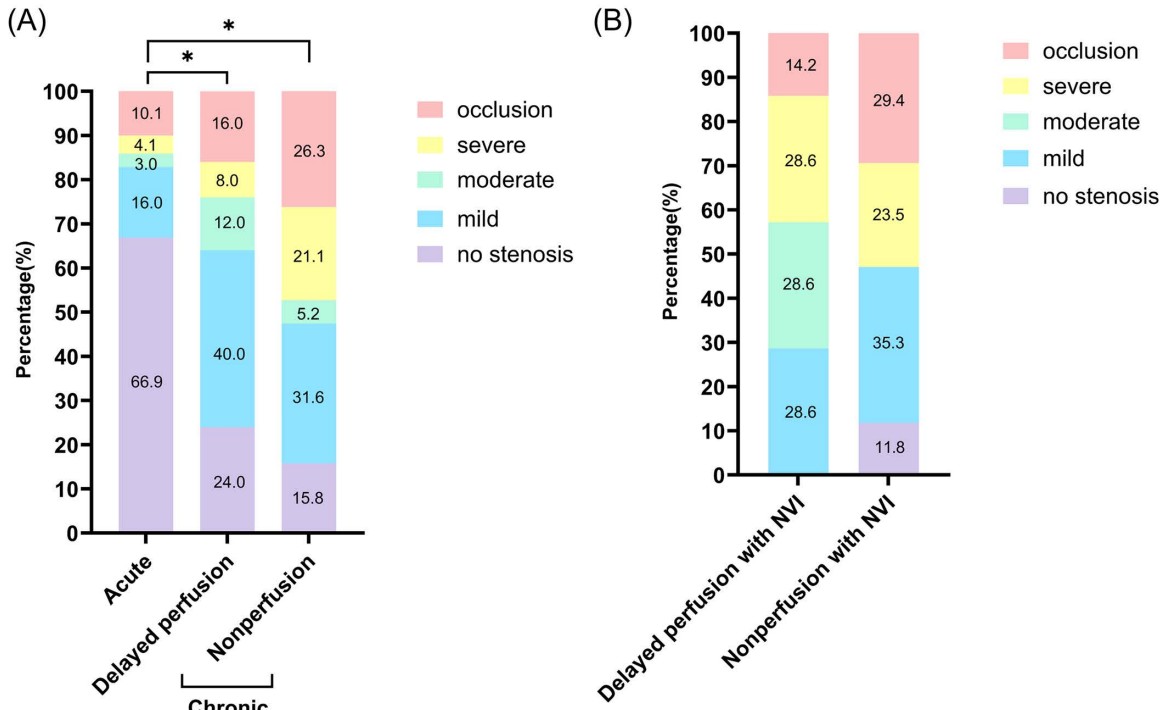

**Fig 5. Comparison of the severity of carotid stenosis according to North American Symptomatic Carotid Endarterectomy Trial(NASCET) Criteria across CRAO subtypes.** Comparison of carotid artery stenosis severity across CRAO subtypes based on NASCET criteria. CRAO classification was determined using fluorescein angiography (FA) performed at 1 month after onset. Carotid stenosis was categorized into five levels: no stenosis, mild, moderate, severe, and occlusion. Cases without carotid stenosis assessment were excluded from this analysis in order to accurately reflect the distribution among assessable patients only. **(A)** Comparison between three CRAO subgroups: acute CRAO, chronic CRAO with delayed perfusion, and chronic CRAO with nonperfusion. **(B)**: Comparison between chronic CRAO patients who developed neovascularization of the iris (NVI): delayed perfusion with NVI vs nonperfusion with NVI. Each bar represents the percentage of patients in each group according to the severity of carotid stenosis. P value was calculated by the chi-square test.

developing within an average of three months [3]. Importantly, that study found that none of the patients who developed iris neovascularization achieved reperfusion on follow-up FA, while 94.7% of those without neovascularization demonstrated at least partial arterial reperfusion (p < 0.001). Similarly, Rudkin et al. reported that 18.2% of CRAO patients developed neovascularization within an average of 8.5 weeks [6], while Duker et al. found an 18% incidence of NVG among 33 CRAO patients [8]. These consistent findings across multiple studies strengthen the evidence for a causal relationship between chronic retinal ischemia following CRAO and the development of anterior segment neovascularization.

The risk factors for developing neovascular complications after CRAO extend beyond retinal perfusion status. Lo et al. identified chronic kidney disease and a history of glaucoma as important risk factors for ocular neovascularization after CRAO, underscoring the systemic and multifactorial nature of these complications [17]. Their findings suggest that patients with coexisting systemic or ocular conditions may require especially close follow-up and earlier intervention to help reduce ischemic damage and neovascular complications, which aligns with our observations regarding the higher prevalence of systemic comorbidities in chronic CRAO patients.

Despite strong evidence linking CRAO to neovascular complications, some studies have reported a lower incidence of NVI and NVG. Hayreh et al. found that only 2.5% of non-arteritic CRAO patients developed NVG, challenging the assumption that CRAO directly leads to neovascular complications [2]. This discrepancy may be attributed to differences in study design, patient selection, and follow-up duration. Hayreh's cohort included transient CRAO and BRAO, reducing

 

the overall incidence of NVG. Additionally, while Hayreh suggested that NVI in CRAO is primarily caused by carotid artery stenosis, our data indicate that even patients with no or mild carotid stenosis developed NVI, suggesting that chronic CRAO itself is an independent risk factor for neovascular complication [2,18].

Another important finding of this study is the close association between chronic CRAO and increased risk of stroke. Our prior study using nationwide data indicates that the risk of stroke is highest immediately after acute CRAO attack emphasizing CRAO as an early and critical marker of systemic vascular risk [12]. A longitudinal study on cohorts of CRAO patients showed that large artery atherosclerosis (LAA) is the most common etiology of retinal artery occlusion and significantly increases the risk of subsequent vascular events (HR 3.94, 95% CI 1.21–12.81) and most of the vascular events was ipsilateral ischemic stroke [11]. We also previously reported that in cosmetic filler-related ophthalmic and retinal artery occlusion, larger size of emboli in the ophthalmic or retinal arteries is associated with both severity of retinal artery occlusion and the risk of cerebral stroke [19]. Taken together, the higher stroke risk observed in chronic CRAO patients could be related to the embolic characteristics of the occlusion: chronic CRAO may involve larger emboli, which is more likely to originate from significant carotid atherosclerosis leading to higher risk of ipsilateral stroke.

The clinical implications of our findings emphasize the importance of serial FA monitoring in identifying high-risk CRAO patients and optimizing management beyond the acute phase. While previous studies have shown the utility of baseline structural assessment for prognosis [20] and the impact of capillary nonperfusion on outcomes [21], our study extends these findings by demonstrating that FA-based perfusion classification beyond the acute phase. While FA is routinely used to evaluate retinal perfusion early after CRAO onset, our study highlights its critical role beyond one month, as persistent nonperfusion strongly predicts an increased risk of neovascular complications. Since high-risk nonperfusion cases may benefit from early prophylactic PRP to prevent NVG, routine FA monitoring within the first 1–3 months facilitates timely intervention and more effective risk stratification. Previous research has demonstrated successful management of CRAO-related neovascular glaucoma with bevacizumab combined with PRP [10,22]. Although guidelines for other ischemic retinal diseases support PRP to reduce NVG risk, specific recommendations for CRAO remain limited. Our findings suggest that similar prophylactic principles may be applicable, particularly in chronic CRAO cases with persistent nonperfusion.

Our study has several important limitations. Survivorship bias represents a significant concern, as patients excluded for lack of follow-up fluorescein angiography or short follow-up duration (n = 281, 49%) may represent cases with different outcomes, potentially affecting the apparent prevalence of chronic CRAO and neovascular complications. Selection bias may also influence our findings, as our tertiary referral center may be enriched for severe cases or patients with complications, potentially overestimating complication rates compared to community practice. Our statistical analysis is limited by the use of univariable comparisons without multivariable modeling. While the strong associations between perfusion status and outcomes (90% NVI rate in nonperfusion cases) suggest robust relationships, the small sample size in the nonperfusion group (n = 19) and high event rate precluded meaningful multivariable regression to avoid overfitting. Future prospective studies with larger sample sizes should incorporate multivariable Cox regression models to identify independent risk factors and develop validated prediction models. Moreover, our carotid stenosis analysis has important limitations. NASCET criteria focus on luminal narrowing but do not capture plaque characteristics such as ulceration or surface irregularity that may influence embolic risk. The observation that neovascular complications developed even in patients with mild or no carotid stenosis suggests that small emboli from minimal plaque disruption, alternative embolic sources, or local retinal vascular factors may contribute to persistent ischemia. Additionally, the lack of intraocular VEGF and inflammatory marker measurements limits our understanding of the underlying mechanisms driving neovascularization in chronic CRAO. Future prospective studies incorporating standardized treatment protocols and biomarker assessments are needed to validate our findings and optimize management strategies for chronic CRAO patients. In addition, further studies are needed to evaluate whether early prophylactic PRP can help prevent NVI in chronic CRAO with nonperfusion. Despite these limitations, this study provides critical insights into the prognostic value of FA in chronic CRAO, demonstrating its effectiveness in classifying cases based on perfusion status and linking this classification to clinical outcomes.

In conclusion, approximately one quarter of acute non-arteritic CRAO patients progress to chronic CRAO, which is associated with a significantly increased risk of stroke, ocular neovascularization and visual impairment, particularly in cases where retinal arterial nonperfusion persist. The strong association between nonperfusion and neovascular complications (with NVI occurring in nearly 90% of nonperfusion cases) underscores the importance of FA monitoring for chronic CRAO 1–3 months after the initial attack to identify high-risk patients. Therefore, a different approach is required for chronic CRAO including close monitoring for neovascular complications and proactive efforts to reduce cerebrovascular risk.

## Author contributions

**Conceptualization:** Jae Ryong Song, Se Joon Woo.

**Data curation:** Jae Ryong Song.

**Formal analysis:** Jae Ryong Song, Se Joon Woo.

**Investigation:** Jae Ryong Song.

**Supervision:** Se Joon Woo.

**Writing – original draft:** Jae Ryong Song.

**Writing – review & editing:** Jae Ryong Song, Se Joon Woo.

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
