## [Decision Letter · Decision Letter 0]

15 Sep 2025

Dear Dr. Woo,

Thank you for submitting your manuscript to PLOS ONE. After careful consideration, we feel that it has merit but does not fully meet PLOS ONE’s publication criteria as it currently stands. Therefore, we invite you to submit a revised version of the manuscript that addresses the points raised during the review process.

We look forward to receiving your revised manuscript.

Kind regards,

Oana Dumitrascu, M.D.

Academic Editor

PLOS ONE

2. In the online submission form, you indicated that [The datasets generated and/or analyzed during the current study contain sensitive patient information and cannot be shared publicly due to patient privacy and confidentiality restrictions under Korean personal information protection laws. The data are available from the corresponding author and the Seoul National University Bundang Hospital Institutional Review Board upon reasonable request and following approval of a data sharing agreement that ensures compliance with ethical and legal requirements for handling confidential medical data.].

Additional Editor Comments:

Reviewer #1:

This is a well-written paper and an intriguing study about a potential new definition/name for the chronic sequelae of CRAO—Chronic CRAO. The paper is well-organized, well-written, and produces some novel findings that are worth publishing. However, I have one large reservation and a few other comments:

1.Chronic CRAO—this conjures ocular ischemic syndrome, and sounds like the intent was to show persistent ischemia after the acute CRAO event. I think the naming of a new entity is a bit of a misnomer—CRAO is by definition a stroke of the central retinal artery and much like cerebral stroke, it is an emergency. We do not want ophthalmologists and ER Docs as well as neurologists to have to differentiate acute CRAO from chronic CRAO when they are working up for stroke, after just getting it in the American Heart Association/American Stroke Association guidelines. In fact, stroke does not have an analogous “chronic stroke” but instead, stroke with chronic sequelae, whether ischemic / hemorrhagic / revealing carotid artery or intracerebral artery critical stenosis.

I wonder if the language could be changed to CRAO with persistent retinal ischemia, or some similar name. It’s not as catchy but it would avoid the chronic label, which suggests that it comes on slowly and persists and isn’t an emergency, rather than the same acute onset and then persistence.

2. I really like the evidence presented and the figures, but I would add a table to show a comparison between the FA perfusion times between acute CRAO, “chronic CRAO” in both delayed and non-perfused groups for more justification of their new definition. I would also show the significant difference in p-values.

3. Fig 2 panel F Shows patchy choroidal non-perfusion that suggest potential arteritis. I would recheck that sample to ensure that giant cell arteritis was effectively ruled out.

4. Fig 3 is great.

5. Fig 4 and 5, would add p-value bars for comparison of different groups to show significant differences.

6. I think figure 6 is very confusing and hard to interpret even with the legend. Consider leaving out or revising to something more familiar.

Reviewer #2:

This is a well-written paper and an intriguing study about a potential new definition/name for the chronic sequelae of CRAO—Chronic CRAO. The paper is well-organized, well-written, and produces some novel findings that are worth publishing. However, I have one large reservation and a few other comments:

1.Chronic CRAO—this conjures ocular ischemic syndrome, and sounds like the intent was to show persistent ischemia after the acute CRAO event. I think the naming of a new entity is a bit of a misnomer—CRAO is by definition a stroke of the central retinal artery and much like cerebral stroke, it is an emergency. We do not want ophthalmologists and ER Docs as well as neurologists to have to differentiate acute CRAO from chronic CRAO when they are working up for stroke, after just getting it in the American Heart Association/American Stroke Association guidelines. In fact, stroke does not have an analogous “chronic stroke” but instead, stroke with chronic sequelae, whether ischemic / hemorrhagic / revealing carotid artery or intracerebral artery critical stenosis.

I wonder if the language could be changed to CRAO with persistent retinal ischemia, or some similar name. It’s not as catchy but it would avoid the chronic label, which suggests that it comes on slowly and persists and isn’t an emergency, rather than the same acute onset and then persistence.

2. I really like the evidence presented and the figures, but I would add a table to show a comparison between the FA perfusion times between acute CRAO, “chronic CRAO” in both delayed and non-perfused groups for more justification of their new definition. I would also show the significant difference in p-values.

3. Fig 2 panel F Shows patchy choroidal non-perfusion that suggest potential arteritis. I would recheck that sample to ensure that giant cell arteritis was effectively ruled out.

4. Fig 3 is great.

5. Fig 4 and 5, would add p-value bars for comparison of different groups to show significant differences.

6. I think figure 6 is very confusing and hard to interpret even with the legend. Consider leaving out or revising to something more familiar.

Reviewers' comments:

Reviewer's Responses to Questions

**Comments to the Author**

1. Is the manuscript technically sound, and do the data support the conclusions?

Reviewer #1: Yes

Reviewer #2: Yes

2. Has the statistical analysis been performed appropriately and rigorously?

Reviewer #1: Yes

Reviewer #2: Yes

3. Have the authors made all data underlying the findings in their manuscript fully available?

Reviewer #1: Yes

Reviewer #2: Yes

4. Is the manuscript presented in an intelligible fashion and written in standard English?

Reviewer #1: Yes

Reviewer #2: Yes

Reviewer #1: This is a well-written paper and an intriguing study about a potential new definition/name for the chronic sequelae of CRAO—Chronic CRAO. The paper is well-organized, well-written, and produces some novel findings that are worth publishing. However, I have one large reservation and a few other comments:

1.Chronic CRAO—this conjures ocular ischemic syndrome, and sounds like the intent was to show persistent ischemia after the acute CRAO event. I think the naming of a new entity is a bit of a misnomer—CRAO is by definition a stroke of the central retinal artery and much like cerebral stroke, it is an emergency. We do not want ophthalmologists and ER Docs as well as neurologists to have to differentiate acute CRAO from chronic CRAO when they are working up for stroke, after just getting it in the American Heart Association/American Stroke Association guidelines. In fact, stroke does not have an analogous “chronic stroke” but instead, stroke with chronic sequelae, whether ischemic / hemorrhagic / revealing carotid artery or intracerebral artery critical stenosis.

I wonder if the language could be changed to CRAO with persistent retinal ischemia, or some similar name. It’s not as catchy but it would avoid the chronic label, which suggests that it comes on slowly and persists and isn’t an emergency, rather than the same acute onset and then persistence.

2. I really like the evidence presented and the figures, but I would add a table to show a comparison between the FA perfusion times between acute CRAO, “chronic CRAO” in both delayed and non-perfused groups for more justification of their new definition. I would also show the significant difference in p-values.

3. Fig 2 panel F Shows patchy choroidal non-perfusion that suggest potential arteritis. I would recheck that sample to ensure that giant cell arteritis was effectively ruled out.

4. Fig 3 is great.

5. Fig 4 and 5, would add p-value bars for comparison of different groups to show significant differences.

6. I think figure 6 is very confusing and hard to interpret even with the legend. Consider leaving out or revising to something more familiar.

Reviewer #2: The introduction does a good job of highlighting CRAO as both an ophthalmologic emergency and a marker of systemic vascular disease. The framing is clinically relevant, and the discussion of neovascular complications is strengthened by linking VEGF to NVI/NVG and drawing parallels to other ischemic retinopathies.

Methods

The methodology is systematic and clearly divided into subsections (Study Population, Clinical Evaluation/Treatment, Follow-up, Systemic Evaluation). However, greater transparency in case confirmation would improve reproducibility.

While you report screening 570 cases and analyzing 289, it is not clear whether diagnoses were confirmed by ICD coding, review of imaging/FA, or adjudication by specialists. A brief clarification here would strengthen confidence in the dataset.

The operational definition of chronic CRAO (>1 month with delayed ART/AVT) is clear, but because this cutoff may differ from prior studies, a short justification or citation would be valuable. Similarly, stroke ascertainment is described as MRI-based, but CT confirmation is not mentioned — was CT excluded or not used? Finally, the definition of “0-month follow-up strokes” (those found on MRI at CRAO onset) is confusing.

Clarifying whether these represent baseline concurrent events or early incident strokes would help readers interpret the results correctly.

Results

The results are comprehensive and flow logically from CRAO subtypes → ocular outcomes → systemic outcomes.

The structure aligns well with the study’s aims. But some repetition exists where numbers are restated both in text and tables (stroke prevalence, NVI rates). Streamlining this by highlighting key takeaways in the text and leaving full detail to the tables would improve the paper .

Stroke outcomes are well reported, but the temporal dimension could be clearer — distinguishing concurrent strokes at baseline from incident strokes during follow-up is essential. It might also help to summarize the overall trend: a progressive worsening from delayed perfusion to nonperfusion, both in terms of ocular complications and systemic vascular risk.

Discussion

The attempt to define chronic CRAO as a novel concept may be overstated. It would be more accurate to frame this as proposing and validating an FA-based classification in your cohort, which also highlights its contribution without risking overinterpretation.

The strong associations between perfusion status and NVI/NVG (90% NVI in the nonperfusion group) are compelling. But, readers will want to know whether these were evaluated with multivariable regression or time-to-event analysis.

If Cox models or other adjusted analyses were used, please summarize them; if not, acknowledging this as a limitation and suggesting prospective validation would strengthen the manuscript.

The interpretation of carotid stenosis also needs nuance. While it is notable that NVI developed even in cases with mild or no stenosis, stenosis assessment was incomplete in some patients, and plaque morphology/vulnerability is not captured by NASCET percentage alone.

reporting that embolic or unstable plaque sources may have contributed would balance the discussion.

The retrospective single-center design introduces important biases. Patients excluded for lack of follow-up FA or short follow-up may represent a survivorship bias, potentially inflating the apparent prevalence of chronic nonperfusion and NVI. Explicitly noting this limitation would improve transparency.

**Do you want your identity to be public for this peer review?** For information about this choice, including consent withdrawal, please see our Privacy Policy

Reviewer #1: No

Reviewer #2: No

---

## [Author Response · Author response to Decision Letter 1]

19 Sep 2025

September 18, 2025

Manuscript ID: PONE-D-25-40463

Title: Chronic Central Retinal Artery Occlusion: Clinical Manifestations, Ocular Neovascular Complications, and Risk of Stroke

Dear Editor and Reviewers,

We sincerely appreciate the thoughtful and constructive feedback from both reviewers. Your comments have significantly improved the quality and clarity of our manuscript. All comments from the reviewers were very thoughtful and have helped us to improve our manuscript. We have carefully checked our manuscript and made appropriate revisions based on the reviewers’ suggestions.

A point-by-point response to the editor’s comments is presented below.

Response to Reviewer #1

Reviewer Comment 1:

"Chronic CRAO—this conjures ocular ischemic syndrome, and sounds like the intent was to show persistent ischemia after the acute CRAO event. I think the naming of a new entity is a bit of a misnomer—CRAO is by definition a stroke of the central retinal artery and much like cerebral stroke, it is an emergency. We do not want ophthalmologists and ER Docs as well as neurologists to have to differentiate acute CRAO from chronic CRAO when they are working up for stroke, after just getting it in the American Heart Association/American Stroke Association guidelines. In fact, stroke does not have an analogous "chronic stroke" but instead, stroke with chronic sequelae, whether ischemic / hemorrhagic / revealing carotid artery or intracerebral artery critical stenosis. I wonder if the language could be changed to CRAO with persistent retinal ischemia, or some similar name. It's not as catchy but it would avoid the chronic label, which suggests that it comes on slowly and persists and isn't an emergency, rather than the same acute onset and then persistence."

Response: We understand and appreciate the reviewer's concern about potential confusion in emergency settings. However, we respectfully propose maintaining the "Chronic CRAO" terminology for the following reasons.

We emphasize that "Chronic CRAO" is used strictly as a descriptive stage label for the post-acute (chronic) phase following an acute CRAO event, not as a new disease entity. This terminology describes patients with FA-confirmed persistent retinal ischemia ≥1 month after acute onset, used solely for prognosis and follow-up risk stratification rather than acute triage decisions.

While "CRAO with persistent retinal ischemia" is semantically accurate, it is considerably longer, less recognizable in clinical communication, and departs from established usage patterns. "Chronic CRAO" provides concise, immediately recognizable risk signaling for clinicians managing long-term sequelae.

"Chronic CRAO" has been used in various peer-reviewed publications, reviews, and imaging studies to describe the late phase and its OCT/OCTA/FA features and sequelae. The following literature is a sample of publications that employ the term "chronic CRAO".

1. Wong CL, Ang M, Tan ACS. Clinical Applications of Optical Coherence Angiography Imaging in Ocular Vascular Diseases. Applied Sciences. 2019;9: 2577. doi:10.3390/app9122577

2. Mehta N, Marco RD, Goldhardt R, Modi Y. Central Retinal Artery Occlusion: Acute Management and Treatment. Curr Ophthalmol Rep. 2017;5: 149–159. doi:10.1007/s40135-017-0135-2

3. Greene DP, Richards CP, Ghazi NG. Comparison of Optical Coherence Tomography Findings in a Patient with Central Retinal Artery Occlusion in One Eye and End-stage Glaucoma in the Fellow Eye. Middle East Afr J Ophthalmol. 2012;19: 247–250. doi:10.4103/0974-9233.95265

To prevent any potential confusion in emergency settings, we have implemented explicit safeguards across our manuscript.:

Line 25-26: This retrospective cohort study investigated clinical characteristics, neovascular complications, and stroke risk in patients with chronic CRAO defined as the post-acute phase state characterized by FA-confirmed persistent retinal ischemia ≥1 month after acute onset.

Line 53-55: While the acute emergency phase of CRAO is the focus of most clinical attention, the post-acute phase presents distinct clinical challenges.

Line 79-80: In this study, we define "Chronic CRAO" as the post-acute phase (≥1 month after onset) characterized by FA-confirmed persistent retinal ischemia, distinct from the acute emergency presentation.

Line 134: “Chronic CRAO”(CRAO with persistent retinal ischemia) was defined as cases where fluorescein angiography performed more than one month after diagnosis showed persistent delay in arm-to-retina time (>20 seconds) and arteriovenous transit time (>11 seconds).

Reviewer Comment 2:

"I really like the evidence presented and the figures, but I would add a table to show a comparison between the FA perfusion times between acute CRAO, "chronic CRAO" in both delayed and non-perfused groups for more justification of their new definition. I would also show the significant difference in p-values."

Response: We have added quantitative comparison of fluorescein angiography perfusion parameters to Table 1, including both initial and follow-up arm-to-retina times to provide objective justification for our classification system

Moreover, The following sentences have been added to the Results section

Line 188-191

“In terms of retinal perfusion parameters, patients with chronic CRAO had longer initial arm-to-retina times (25.0 ± 9.2 vs. 20.3 ± 10.0 seconds, p = 0.003) and follow-up arm-to-retina times (23.3 ± 6.5 vs. 18.0 ± 6.4 seconds, p < 0.001) compared to acute CRAO, indicating persistent impaired retinal circulation.”

Reviewer Comment 3:

"Fig 2 panel F Shows patchy choroidal non-perfusion that suggest potential arteritis. I would recheck that sample to ensure that giant cell arteritis was effectively ruled out."

Response 3: We carefully re-evaluated this case to exclude the possibility of arteritic CRAO. The patient was a 68-year-old male without systemic symptoms suggestive of giant cell arteritis (such as headache, scalp tenderness, or jaw claudication). On examination, there was no temporal artery tenderness. Laboratory results showed normal CRP and platelet count. Moreover, the overall clinical course was typical of non-arteritic CRAO. These features strongly argue against arteritic CRAO. Nevertheless, to prevent potential misinterpretation by readers, we have replaced the original image (Fig 2 D-F) with another representative non-arteritic CRAO case showing similar angiographic features.

Reviewer Comment 4:

"Fig 3 is great."

Response 4: We appreciate the positive feedback on Figure 3.

Reviewer Comment 5:

"Fig 4 and 5, would add p-value bars for comparison of different groups to show significant differences."

Response 5: We have enhanced both figures with statistical comparison bars and p-values. Overall stroke incidence and previous stroke were significantly frequent in both type of chronic CRAO than acute CRAO. Moreover, statistical annotations demonstrate that both chronic CRAO subtypes show significantly worse carotid stenosis distributions compared to acute CRAO. While the nonperfusion group showed a trend toward higher proportions of severe stenosis and occlusion, the overall distributions between the two chronic subtypes were not statistically different,

Reviewer Comment 6:

"I think figure 6 is very confusing and hard to interpret even with the legend. Consider leaving out or revising to something more familiar."

Response: We agree that Figure 6 was unnecessarily complex and potentially confusing for the main manuscript. To avoid any confusion and to improve clarity, we have removed the Sankey diagram (Figure 6) from the revised manuscript.

Response to Reviewer #2

Reviewer #2: The introduction does a good job of highlighting CRAO as both an ophthalmologic emergency and a marker of systemic vascular disease. The framing is clinically relevant, and the discussion of neovascular complications is strengthened by linking VEGF to NVI/NVG and drawing parallels to other ischemic retinopathies.

Methods

The methodology is systematic and clearly divided into subsections (Study Population, Clinical Evaluation/Treatment, Follow-up, Systemic Evaluation). However, greater transparency in case confirmation would improve reproducibility.

While you report screening 570 cases and analyzing 289, it is not clear whether diagnoses were confirmed by ICD coding, review of imaging/FA, or adjudication by specialists. A brief clarification here would strengthen confidence in the dataset.

Response: The following paragraphs have been added to the Methods section

Line 97-102

“Initial screening involved electronic medical record review using ICD-10 codes (H34.10-H34.13), followed by clinical validation where all cases were reviewed by board-certified retinal specialists (S.J.W. and J.R.S.). Imaging confirmation required systematic review of fundus photography confirming cherry-red spot and retinal whitening, fluorescein angiography demonstrating retinal arterial occlusion, and OCT confirming retinal structural changes consistent with CRAO.”

The operational definition of chronic CRAO (>1 month with delayed ART/AVT) is clear, but because this cutoff may differ from prior studies, a short justification or citation would be valuable.

Response: The following sentences have been added to the Methods section

Line 137-140

The 1-month cutoff was chosen based on established literature showing that most acute retinal changes in CRAO resolve within 2-4 weeks, with persistent findings beyond this time point representing chronic ischemic sequelae rather than ongoing acute changes.[14] This timeframe also aligns with clinical follow-up protocols.

Similarly, stroke ascertainment is described as MRI-based, but CT confirmation is not mentioned — was CT excluded or not used?

Response: In our study, stroke ascertainment was based on both MRI and CT imaging, depending on clinical circumstances and timing of presentation. We will clarify this in the revised Methods section as follows.

Line 158-159

“Stroke events were assessed using brain imaging (MRI and/or CT) performed at the time of CRAO diagnosis or during follow-up.”

Finally, the definition of “0-month follow-up strokes” (those found on MRI at CRAO onset) is confusing. Clarifying whether these represent baseline concurrent events or early incident strokes would help readers interpret the results correctly.

Response: We have clarified this important distinction by implementing a comprehensive three-category stroke classification system in the Methods section.

Line 159-164

We classified stroke events into three distinct categories based on temporal relationships to CRAO onset. Previous stroke was defined by patient history or old infarction on brain MRI. Concurrent stroke represented acute cerebral infarctions detected on MRI performed at the time of CRAO diagnosis, indicating simultaneous vascular events within the same or different vascular territories. Incident stroke included new cerebral infarctions detected on follow-up imaging performed after CRAO diagnosis.

Results

The results are comprehensive and flow logically from CRAO subtypes → ocular outcomes → systemic outcomes.

The structure aligns well with the study’s aims. But some repetition exists where numbers are restated both in text and tables (stroke prevalence, NVI rates). Streamlining this by highlighting key takeaways in the text and leaving full detail to the tables would improve the paper .

Stroke outcomes are well reported, but the temporal dimension could be clearer — distinguishing concurrent strokes at baseline from incident strokes during follow-up is essential. It might also help to summarize the overall trend: a progressive worsening from delayed perfusion to nonperfusion, both in terms of ocular complications and systemic vascular risk.

Response:

We have comprehensively streamlined the Results section by focusing the text on key findings, trends, and clinical significance rather than repeating specific percentages already presented in tables. This approach eliminates redundancy while enhancing the clinical narrative.

We also revised our results about stroke as below.

Line 242-249

The proportion of patients with previous stroke was significantly higher in the chronic CRAO group compared to the acute CRAO group. Specifically, previous stroke increased from 26.5% in acute CRAO to 40.0% in delayed perfusion (p = 0.0498) and 52.6% in nonperfusion cases (p = 0.0158 vs. acute CRAO). When considering overall stroke burden (including previous, concurrent, and incident stroke), there was a significant stepwise increase across CRAO types: 34.0% in acute CRAO, 54.5% in delayed perfusion (p = 0.005 vs. acute), and 63.2% in nonperfusion cases (p = 0.011 vs. acute). Concurrent and incident stroke events also showed a stepwise increase across groups, though these differences were not statistically significant. (Table 1, Fig 4)

Discussion

The attempt to define chronic CRAO as a novel concept may be overstated. It would be more accurate to frame this as proposing and validating an FA-based classification in your cohort, which also highlights its contribution without risking overinterpretation.

The strong associations between perfusion status and NVI/NVG (90% NVI in the nonperfusion group) are compelling. But, readers will want to know whether these were evaluated with multivariable regression or time-to-event analysis. If Cox models or other adjusted analyses were used, please summarize them; if not, acknowledging this as a limitation and suggesting prospective validation would strengthen the manuscript.

The interpretation of carotid stenosis also needs nuance. While it is notable that NVI developed even in cases with mild or no stenosis, stenosis assessment was incomplete in some patients, and plaque morphology/vulnerability is not captured by NASCET percentage alone. reporting that embolic or unstable plaque sources may have contributed would balance the discussion.

The retrospective single-center design introduces important biases. Patients excluded for lack of follow-up FA or short follow-up may represent a survivorship bias, potentially inflating the apparent prevalence of chronic nonperfusion and NVI. Explicitly noting this limitation would improve transparency.

Response: The following paragraphs have been added to the Discussion section

Line 361-377

Our study has several important limitations. Survivorship bias represents a significant concern, as patients excluded for lack of follow-up fluorescein angiography or short follow-up duration (n=281, 49%) may represent cases with different outcomes, potentially affecting the apparent prevalence of chronic CRAO and neovascular complications. Selection bias may also influence our findings, as our tertiary referral center may be enriched for severe cases or patients with complications, potentially overestimating complication rates compared to community practice. Our statistical analysis is limited by the use of univariable comparisons without multivariable modeling. While the strong associations between perfusion status and outcomes (90% NVI rate in nonperfusion cases) suggest robust relationships, the small sample size in the nonperfusion group (n=19) and high event rate precluded meaningful multivariable regression to avoid overfitting. Future prospective studies with larger sample sizes should incorporate multivariable Cox regression models to identify independent risk factors and develop validated prediction models. Moreover, our carotid stenosis analysis has important limitations. NASCET criteria focus on luminal narrowing but do not capture plaque characteristics such as ulceration or surface irregularity that may influence embolic risk. The observation that neovascular complications developed even in patients with mild or no carotid stenosis suggests that small emboli from minimal plaque disruption, alternative embolic sources, or local retinal vascular factors may contribute to persistent ischemia.

Thank you for your detailed feedback and consideration of our submitted paper. Your feedback has helped us address important issues and significantly improve our manuscript. We believe these comprehensive revisions have enhanced the scientific rigor, clarity, and clinical relevance of our work. We hope our

---

## [Decision Letter · Decision Letter 1]

6 Oct 2025

Chronic Central Retinal Artery Occlusion: Clinical Manifestations, Ocular Neovascular Complications, and Risk of Stroke

PONE-D-25-40463R1

Dear Dr. Woo,

We’re pleased to inform you that your manuscript has been judged scientifically suitable for publication and will be formally accepted for publication once it meets all outstanding technical requirements.

Kind regards,

Oana Dumitrascu, M.D.

Academic Editor

PLOS ONE

Additional Editor Comments (optional):

The revisions are acceptable for publication in current format.

Reviewers' comments:

Reviewer's Responses to Questions

**Comments to the Author**

Reviewer #1: All comments have been addressed

Reviewer #3: All comments have been addressed

2. Is the manuscript technically sound, and do the data support the conclusions?

Reviewer #1: Partly

Reviewer #3: Yes

3. Has the statistical analysis been performed appropriately and rigorously?

Reviewer #1: Yes

Reviewer #3: Yes

4. Have the authors made all data underlying the findings in their manuscript fully available?

Reviewer #1: Yes

Reviewer #3: Yes

5. Is the manuscript presented in an intelligible fashion and written in standard English?

Reviewer #1: Yes

Reviewer #3: Yes

Reviewer #1: I still have reservations about the use of the term Chronic CRAO but the paper is otherwise well done.

Reviewer #3: all comments addressed well and also revised Version in modified in the method, result and discussion section .

**Do you want your identity to be public for this peer review?** For information about this choice, including consent withdrawal, please see our Privacy Policy

Reviewer #1: No

Reviewer #3: No

---

## [Editor Report · Acceptance letter]

PONE-D-25-40463R1

PLOS ONE

Dear Dr. Woo,

I'm pleased to inform you that your manuscript has been deemed suitable for publication in PLOS ONE. Congratulations! Your manuscript is now being handed over to our production team.

Kind regards,

on behalf of

Dr. Oana Dumitrascu

Academic Editor

PLOS ONE